# Effect of Hydrophilic Polyurethane on Interfacial Shear Strength of Pisha Sandstone Consolidation under Freeze–Thaw Cycles

**DOI:** 10.3390/polym15092131

**Published:** 2023-04-29

**Authors:** Wenbo Ma, Ke Yang, Xuan Zhou, Zhengdong Luo, Yuefei Guo

**Affiliations:** 1School of Mechanical Engineering and Mechanics, Xiangtan University, Xiangtan 411105, China; 2School of Civil Engineering, Xiangtan University, Xiangtan 411105, China; 3Yueyang Road and Bridges Base Construction General Company, Yueyang 414000, China

**Keywords:** W-OH, Pisha sandstone, freeze–thaw cycles, interface strength, ethylene vinyl acetate (EVA)

## Abstract

The W-OH type polyurethane (W-OH) has been proven to be an economical and environmentally friendly slope protection solution for slope maintenance in Pisha sandstone areas. However, the Pisha area belongs to a typical temperate continental climate with large diurnal temperature changes in winter, spring, and autumn and freezing and thawing occurring alternately between days and nights. Under freeze–thaw cycle conditions, the effect of slope treatment largely depends on the interface shear strength between W-OH-treated Pisha sandstone and pristine sandstone. Therefore, this paper studies the interfacial shear strength and long-term durability of Bisha sandstone consolidation (W-OH-treated Pisha sandstone) and Pisha sandstone under freeze–thaw cycles. First, the effects of different W-OH concentrations and different water contents on the freeze–thaw cycle interface were studied using a direct shear test. Based on the experimental results, the W-OH material was further modified with ethylene vinyl acetate (EVA). Finally, the damaged surface of the sample was observed through an ultra-depth-of-field microscope, and the damage mechanism of the interface caused by the freeze–thaw cycles was further discussed. The experimental results show that the peak shear strength at the interface increases with the increase in W-OH concentration and decreases with the increase in freeze–thaw cycles. The cohesion at the interface generally increases with the increase in W-OH concentration and reaches a maximum value of 43.6 kPa when the W-OH concentration is 10%. At the same time, under the condition of high water content, the curing of the W-OH material has no significant effect on the bonding performance of the interface. Using EVA to modify the W-OH material can improve the freeze–thaw durability of the interface. After modification, the interfacial cohesion of the sample increases with the increase in the EVA concentration and can reach 162% of the original. Using an ultra-depth-of-field microscope, it was found that the repeated solidification–melting action of water between the interfaces makes the consolidated body on the damaged surface fall off, resulting in cracks. As the water content between the interfaces increases, the damage to the material is greater. However, the addition of EVA can fill the uncovered pores of W-OH cement, thereby improving the cohesion at the interface and effectively alleviating the freeze–thaw damage caused by the high water content at the interface. The results of this study can provide some theoretical references for slope treatment in the Pisha sandstone area using W-OH materials.

## 1. Introduction

Pisha sandstone is a loose rock series composed of sandstone, sand shale, and argillaceous sandstone. It is widely distributed in Shanxi province and the internal Mongolia Autonomous Region of the Yellow River Basin in China, covering an area of about 16.7 × 10^3^ km^2^ [1,2,3]. Its main mineral composition is quartz (70.5 ± 1.0%), feldspar (16.4 ± 0.5%), muscovite (12.9 ± 1.1%), and hematite (0.2 ± 0.05%) [3]. The Pisha sandstone’s cementitious material, i.e., carbonate mineral calcite, easily dissolves in water. Therefore, it is tough when dry, with a compressive strength of about 2–3 MPa but would collapse when immersed in water. Pisha sandstone [4,5] has a low degree of diagenesis and poor cementation; this special lithology makes Pisha sandstone extremely sensitive to water and leads to very high soil erosion rates (over 20,000 t/(km^2^·a) which frequently occur during and after a rainstorm. These conditions seriously threaten the stability of the slope. The freeze–thaw cycle also affects the stability of the Pisha sandstone and its slope. Therefore, it is necessary to consider the influence of the freeze–thaw cycle on the stability of the Pisha sandstone slope in practical engineering. In this paper, we mainly use weathered particles on the surface of the Pisha sandstone which collapse in water and focus on the nature of the soil; hence, we emphasize the review of research progress dominated by soil studies.

In recent decades, with the influence of global warming, soil freezing and thawing in seasonal permafrost regions have intensified [6]. Repeated frost heave and thawing processes significantly impact the soil’s physical and chemical properties [7], such as soil shear strength [8], soil structure [9], and soil void [10], which affect the stability of the slope [11]. This process will also cause a change in soil water content, change the soil matrix suction, reduce the shear strength, and threaten the stability of the slope [12,13]. Tian et al. [14] conducted porosity measurements, penetration tests, and direct shear tests on the remolded samples of silty clay particles and sand gravel particles extracted from the original soil. The results indicate that long-term internal erosion will lead to a decrease in soil shear strength and further aggravate erosion. Xie et al. [15] found that soil cohesion and uniaxial compressive strength decreased with increased volume and porosity after the freeze–thaw cycle. Ma et al. [16] carried out triaxial shear experiments to study the effects of freeze–thaw cycles on lime soil’s shear strength. It was found that the more freeze–thaw processes, the more pronounced the strength attenuation. These studies show that freezing and thawing will significantly impact the physical properties of soil, so it is necessary to improve the frost resistance of soil through engineering methods.

Engineering methods to improve soil frost resistance mainly include chemical and physical processes. Chemical processes involve adding cement [17], lime [18,19], and other materials to the soil. Physical methods mainly improve the frost resistance of soil through geogrids and geotextiles [20]. In these methods, polyurethane has the advantages of low viscosity, low density, short gelation time, high strength, chemical inertia after hardening, etc. Therefore, polyurethane is considered a stable material that improves soil properties [21]. Liang et al. [22] used modified hydrophilic polyurethane (W-OH) to strengthen weathered Pisha sandstone. It is found that the consolidation of Pisha sandstone can significantly improve corrosion resistance and water retention and effectively alleviate the slope deformation caused by dry and wet and gravity. Yao et al. [23] found that W-OH has proper permeability on the surface of Pisha sandstone. After the W-OH treatment, the sand production of the slope is reduced from 90% to 99%. Ma et al. [24] studied the mechanical properties of W-OH/Pisha sandstone under freeze–thaw cycles. It was found that W-OH consolidation can significantly improve the long-term performance of Pisha sandstone under the freeze–thaw process, and its freeze–thaw resistance increases with the increase in W-OH concentration. The above studies show that the modified hydrophilic polyurethane material (W-OH) has a good effect on the solidification and soil conservation of the Pisha sandstone slope.

However, when the consolidation time of the W-OH solution is considered, the solution will have a certain infiltration depth on the slope [22,23]. The interface between Pisha sandstone consolidation and the Pisha sandstone itself is formed on the hill. Due to the physical and mechanical differences between Pisha sandstone consolidation [25] and Pisha sandstone [26], interfacial shear strength is significant for the stability of the Pisha sandstone slope.

There are few reports on the interfacial shear properties between Pisha sandstone consolidation and Pisha sandstone at home and abroad [24]. Therefore, this paper will study the interfacial shear strength and long-term durability of Bisha sandstone consolidation (W-OH-treated Bisha sandstone) and Bisha sandstone under freeze–thaw cycles. The effects of different W-OH concentrations and different water contents on the freeze–thaw cycle interface were studied using direct shear experiments. According to the experimental results, the W-OH material was further modified with ethylene vinyl acetate (EVA). Finally, the damaged surface of the sample was observed through an ultra-depth microscope, and the damage mechanism of the interface caused by freeze–thaw cycles was further discussed. It was then evaluated in terms of W-OH concentration, initial water content, normal stress, and several freeze–thaw cycles. The success of this study will provide a theoretical basis for slope management in the Pisha sandstone area.

## 2. Materials and Methods

### 2.1. Materials

The Pisha sandstone samples were taken from the Erlaohugou Basin in the Huangfuchuan Basin, a primary tributary of the Yellow River, where the weathering degree of the Pisha sandstone is severe. The Pisha sandstone taken in this test is gray–white. It has a loose structure and suffers from severe water collapse (Figure 1). It belongs to the seriously weathered soft sandstone category. Its basic physical properties are shown in Table 1. Figure 1 shows the immersion of the Pisha sandstone in water over time; it can be seen that the Pisha sandstone quickly disintegrated with the increase in immersion time in the water. The Pisha sandstone particles were passed through the No. 4 sieve (2.36 mm), and the screened Pisha sandstone particles were dried in a drying box at 105 °C for 24 h, as shown in Figure 2a.

The new polyurethane composite (W-OH) is a light yellow to brown oily substance with a density of 1.18 g/cm^3^ (Figure 2a) and free of harmful ingredients such as heavy metals. The material uses water as a curing agent, reacts quickly with water to release CO_2_ gas, forms a porous flexible gel with a particular chemical strength, and effectively permeates into the weathered Pisha sandstone particles with water. After solidification, the loose soil can be connected into a large network structure with a water retention effect (Figure 2b). It plays a good consolidation role and is a new and effective soil and water conservation material.

### 2.2. Specimen Preparation

First, according to the “*Standard for Geotechnical Testing Method*” (GB/T50123-2019), the dry soil sample is crushed and screened by 2 mm to measure its moisture content. Before sample preparation, the soil sample is weighed and placed into a prepared plate. A certain amount of water should be evenly sprayed onto the surface of the soil sample with a small sprayer while pouring and stirring. Finally, the prepared soil samples are evenly spread on a tray, sealed with plastic film, and left for more than 24 h to obtain Pisha sandstone particles with different water content. Then, ring-cut soil samples (61.8 mm in diameter and 20 mm in height) with varying water contents are prepared, as shown in Figure 3a. Three different concentrations of the W-OH solution (6%, 8%, and 10%) were selected based on field practice [25]. The W-OH solution and Pisha sandstone particles were uniformly mixed with a handheld mixer for 1 min. Before solidifying the mixture, it was quickly compacted into the ring knife, and the surface of the ring knife was smoothed out as shown in Figure 3a. Finally, the interface sample is demolded after being stored in the moisturizing bag for 72 h displayed in Figure 3b.

Ethylene vinyl acetate (EVA) is utilized to enhance the dry–wet resistance of the interface sample based on the former study. Based on the published data in reference, two concentrations of 5% and 10% were selected. Sample combinations for all tests are listed in Table 2.

### 2.3. Freeze–Thaw Cycling

The freeze–thaw cycle temperature of the sample was −20 °C~20 °C [24]. The model was frozen in an environmental chamber at −20 °C for 12 h and thawed at 20 °C for 12 h. The process was repeated within the selected cycles (0, 4, 8, 16, 25, 37, 58, and 85). After the freeze–thaw cycle, the samples were placed at room temperature (23 °C) to test the effects of the freeze–thaw process on the interfacial strength.

### 2.4. Testing Procedure

All the tests were carried out with the ZJ quadruple strain control direct shear instrument produced by Nanjing Soil Instrument Factory (Nanjing, China). Figure 4 shows the schematic diagram of the specimen direct shear device. In the testing process, the lower part of the shear box adopted the constant shear displacement rate of 0.8 mm/min. It is worth noting that the shear displacement is applied to four elements simultaneously, and the shear force is measured at each unit. Due to equipment limitations, the vertical displacement during shearing is not counted. The normal stress applied to the specimen is 50 kPa, 100 kPa, 200 kPa, and 300 kPa, respectively. All possible combinations of the test parameters, including moisture content (8%, 12%, 16%, and 20%), W-OH concentration (6%, 8%, and 10%), EVA concentration (5% and 10%), freeze–thaw cycles (0, 4, 8, 16, 25, 37, 58 and 85), and normal stress (50 kPa, 100 kPa, 200 kPa) were evaluated. All possible combinations of test parameters, including moisture content (8%, 12%, 16%, and 20%), W-OH concentration (6%, 8%, and 10%), EVA concentration (5% and 10%), freeze–thaw cycles (0, 4, 8, 16, 25, 37, 58 and 85), and normal stress $\sigma_{n}$ (50 kPa, 100 kPa, 200 kPa) are listed in Table 3.

## 3. Experimental Results

### 3.1. Shear Stress-Displacement

Figure 5 compares the interfacial shear strength of water content $\omega_{i}$ = 12% and normal pressure $\sigma_{n}$ = 50 kPa under different freeze–thaw cycles and W-OH concentrations. Figure 5 shows that the number of freeze–thaw cycles have a specific effect on the peak and residual stress, and a noticeable trend can also be observed in the figure. For example, as shown in Figure 5a, the interfacial shear strength decreases with freeze–thaw cycles and increases when the concentration of W-OH is 6%. In Figure 5b,c, a similar phenomenon can be observed for 8% and 10% W-OH concentrations. The comparison of Figure 5a–c shows that the application of W-OH has a noticeable effect on peak shear strength at the interface but has little or no impact on the residual power. Specifically, under the above moisture content and normal pressure, peak shear strength at the interface increases with the increase in W-OH concentration, and it decreases with the rise in freeze–thaw cycles.

Figure 6 compares the interfacial shear strength of different initial water contents when the W-OH concentration is 6% and $\sigma_{n}=100\;\mathrm{kPa}$. It was found that the initial water content has the same effect on peak shear strength but no obvious effect on the residual strength. Under the specific freeze–thaw cycles, peak shear strength decreases with the increase in initial water content. Figure 7 compares the effects of different normal pressures on peak and residual strengths under specific freeze–thaw cycles when the W-OH concentration is 8%, and the initial water content is 20%. The peak value and residual stress increase with the increase in normal stress. In Figure 7c, most of the freeze–thaw cycle curves show strain hardening under $\sigma_{n}=200\;\mathrm{kPa}$.

### 3.2. Peak and Residual Strength

Figure 8 depicts the relationship between water content, W-OH concentration, peak intensity, and freeze–thaw cycles under constant normal pressure. As can be seen in Figure 8a,b, for the same water content and normal pressure, peak strength generally increases with the increase in W-OH concentration. This is due to the formation of porous flexible cement with specific chemical strength when W-OH is mixed with water and effectively permeates into the weathered Pisha sandstone particles with water. After solidification, the loose soil can be connected to form an extensive network structure with the effect of soil consolidation. With the increase in W-OH concentration, the consolidation effect is more pronounced. The development of initial water content on peak strength has a similar trend which shows that peak strength decreases with the increase in initial water content. Similarly, the effect of freeze–thaw cycles on peak strength has the same trend as the initial water content. The comparison in Figure 8a,b shows that peak strength increases significantly with the increase in normal pressure at the same initial water content and W-OH concentration, which means that friction clearly affects the resistance at the interface.

Figure 9a,b describes the relationship between residual strength and the number of freeze–thaw cycles under $\sigma_{n}$ = 50 kPa and $\sigma_{n}$ = 200 kPa, for all initial water contents and W-OH concentrations, respectively. It is worth noting that under the conditions, part of the shear stress–strain curve is hardened, and there is no residual strength, as shown in Figure 9b. Although the residual strength is less than the peak strength, the comparison in Figure 9a,b shows that the effect of $\sigma_{n}$ on the residual strength is more obvious than the peak strength. In addition, the development law between residual strength and W-OH concentration, initial moisture content, and freeze–thaw cycles in Figure 9 is still similar to that in Figure 8.

Peak shear strength was further studied using the Mohr–Coulomb shear failure criterion:(1)$$\tau_{p}=c_{p}+\sigma_{n}\tan\emptyset_{p},$$
where $c_{p}$ is the cohesion at the interface and $\emptyset_{p}$ is the angle of internal friction at the interface.

Figure 10 shows the variation in cohesion and freeze–thaw cycles at the interface under all moisture contents and W-OH concentrations. In general, $c_{p}$ increases with the increase in W-OH concentration, and this trend is consistent with the above $\tau_{p}$ (peak intensity) and W-OH concentration. When the concentration of W-OH is 6%, the maximum value of $c_{p}$ is 27.9 kPa. When the concentration of W-OH is 10%, the maximum value of $c_{p}$ is 43.6 kPa. From the comparison of the maximum values, the low concentration of W-OH solution did not form an obvious binding force at the interface. In addition, the number of cycles has a significant effect on the $c_{p}.$ In the samples with different moisture contents, the cohesion at the interface decreases with the increase in freeze–thaw cycles. The number of specific freeze–thaw cycles decrease with the addition of moisture content for the same W-OH concentration and different moisture content. Under the condition of low moisture content (8% and 12%), the effect of moisture content on interfacial cohesion is minimal; under the condition of high moisture content (16% and 20%), compared with the state of low moisture content, the effect of water content on interfacial cohesion is significant. Especially when the moisture content is 20%, the role of hydrophilic polyurethane at the interface is minimal. As shown in Figure 10, when the W-OH concentration is 10%, and the corresponding moisture content is 8%, 12%, 16%, and 20% after 85 freeze–thaw cycles, $c_{p}$ decreases from 43.1 kPa, 43.6 kPa, and 18.9 kPa to 5.1 kPa, 12.4 kPa, 7.5 kPa, and 2.3 kPa, respectively.

Figure 11 shows the variation in the internal friction angle and the number of freeze–thaw cycles at the interface under water content and W-OH concentration conditions. As shown in Figure 11a, the maximum and minimum internal friction angles are 28° and 24°, respectively; in Figure 11b, the maximum and minimum internal friction angles are 21° and 32°, respectively; and in Figure 11c, the maximum and minimum internal friction angles are 26° and 32°, respectively. The number of cycles, the concentration of W-OH, and the water content have no significant correlation with the friction angle at the interface.

### 3.3. Improved Interface Freeze–Thaw Cycle Durability

It was found above that the interface was destroyed after 85 cycles when the concentration of W-OH was 6% (Figure 12). For high concentrations of W-OH (8% and 10%) after 85 freeze–thaw cycles, although the interface surface performed well, the interfacial cohesion decreased significantly. Based on the study of the cohesive force between W-OH and concrete and W-OH modified sandstone, the results show that the addition of EVA can enhance not only the adhesion between W-OH and concrete but also the anti-freezing durability of Pisha sandstone. This solves the problem of the long-term durability of the slope under the freeze–thaw cycle not being maintained using W-OH material alone. Therefore, EVA modified W-OH to improve the durability of W-OH, thus protecting sandstone from freeze–thaw cycles.

EVA modified the W-OH solution with a 6% concentration. Figure 13a shows the change in cohesion of modified samples after 0 freeze–thaw cycles. The interfacial cohesion of limited pieces increases with the increase in EVA concentration. This trend is evident for low water content (8% and 12%) but not for high water content (16% and 20%): this is a phenomenon similar to the one described above. When the concentration of W-OH was 6% before modification, the cohesion of the interface was 27.5 kPa, 27.3 kPa, 18.6 kPa, and 13.7 kPa, respectively. The interfacial cohesion of the samples with 10% EVA was 44.5 kPa, 39.7 kPa, 26.7 kPa, and 20.34 kPa, respectively. Figure 13b shows the cohesion ratio between the modified specimen after 85 cycles and the actual model (as shown in Figure 12, when the concentration of W-OH is 6%, the model undergoes 85 processes of freeze–thaw, and the interface is destroyed). After 85 cycles of the freeze–thaw cycle, the interfacial adhesion of the modified samples was good and higher than that of the unmodified models after 58 cycles of water content.

It was found above that a 6% W-OH solution with EVA modification of 10% is relatively complete after 85 freeze–thaw cycles (Figure 14). It can be seen that the durability of the W-OH material freeze–thaw cycle can be further improved by adding EVA.

### 3.4. Research on Damage of Failure Surface after Freeze–Thaw Cycle

To better understand the damaging effect of freeze–thaw cycles on the interface, an ultra-depth-of-field three-dimensional microscope system was used to observe the micro-morphology of the damaged surface after shearing. As shown in the figure, W-OH mixed with water will produce a particular chemical strength of the flexible gel, with water effectively infiltrated into the weathering of sandstone particles. The loose soil, after solidification, can be connected into a large network structure with soil consolidation. It was found that part of the W-OH solid was connected to the Pisha sandstone at the failure surface. Compared to Figure 15a,c, the surface of Figure 15c is rougher, the pores are more prominent, and the curing effect of the material is not apparent. The water swelling of the W-OH polymer is due to the hydrophilic unit in the polyether chain, which makes it highly absorbent. After water absorption, the structure of the polyether chain changes from highly entwining to stretching (Figure 16), thus resulting in the swelling phenomenon. The results show that the interfacial shear strength of the W-OH polymer is not apparent at high water content (16% and 20%). As shown in Figure 15b,d after 58 cycles of freezing and thawing, the solid mass on the failure surface falls off and cracks occur due to the repeated freezing–thawing action of the water between the interfaces. With the increased water content between the interfaces, the freeze–thaw action is obvious, and the damage to the materials is more serious. The large area falling off the tight body can be found in Figure 15d.

Figure 17 is a local micrograph of the shear failure interface of the W-OH-modified material. It was found that the addition of EVA can better fill the uncovered pores of the W-OH cementitious body, thus improving the interface’s cohesion and effectively alleviating the freeze–thaw damage caused by high water content at the interface. As shown in Figure 17c,d, only part of the polymer particles on the surface are shed without apparent cracks.

As shown in Figure 18, further examination of the three-dimensional structure of the failure surface is carried out to study the interface damage mechanism. After 58 cycles of freeze–thaw, the failure surface of W-OH has a significant height difference before modification. The failure surface elevation difference in Figure 18a is 826.9 μm; and the failure surface elevation difference in Figure 18c is 1123.7 μm. It was found that the failure surface became very rough, and the deterioration of W-OH consolidation also caused surface roughness due to freezing, thawing, and the degradation of the cementation effect. The damage increases with the increase in water content between the interfaces. The maximum height difference was 633.4 μm. The surface of W-OH modified by 10% EVA remained relatively flat after freeze–thaw cycles, as shown in Figure 18b,d. The above phenomena prove that the uncovered pores of W-OH material can be filled more adequately after the addition of EVA material to improve interfacial cohesion and durability. These results further support the improvement of freeze–thaw durability of polyurethane-treated sandstone through EVA modification.

## 4. Discussion

As mentioned in the introduction, repeated frost heave and thawing processes will significantly impact the soil’s physical properties, including soil shear strength, compressive strength, and soil porosity. Previous studies mainly focused on improving the durability of the soil’s freeze–thaw cycle. This paper aims to study the effects of the freeze–thaw process on the interfacial shear strength between W-OH consolidation and Pisha sandstone. In this paper, the shear strength of the sample interface increases with the increase in W-OH concentration. At low water contents (8% and 12%), the effect of W-OH material on the sample interface is higher than that of water content at the interface; at high water contents (18% and 20%), the effect of the W-OH material on the sample interface will be weakened with the increase in water content. In addition, a suitable test scheme was adopted to further improve the freeze–thaw resistance of polyurethane-stabilized soil. The results show that after 85 freeze–thaw cycles, the interface of 6% W-OH samples modified by 5% and 10% EVA remains intact, and the interfacial shear strength is higher than that of 58-fold freeze–thaw models without modification.

The results show that the shear strength of the interface increases with the concentration of W-OH because when W-OH is mixed with water, a porous flexible gel with specific chemical strength is formed and effectively stops the water from weathering the sandstone particles. The loose soil after solidification can be connected into a large network structure with soil consolidation, and with the increase in concentration of W-OH, the effect is more pronounced. This result is consistent with previous studies [22,25]. In addition, the freeze–thaw action can be equivalent to the energy absorption and release in the soil, which has a severe and complex effect on soil erosion [27]. The frost heave of soil is very common in freeze–thaw cycles; this can lead to the structural destruction of the earth [28,29]. In this study, soil cohesion decreased with the increase in freeze–thaw processes, dropped slowly at 37 cycles, and reached the minimum at almost 85 cycles. This was an indication that the freeze–thaw cycle had changed the arrangement and combination of soil particles, resulting in a decrease in the structural strength of the soil [30]. In addition, the constant volume change in soil water content during the freezing and thawing process has a specific effect on soil structure [31]. However, at low soil moisture content, the void volume of the soil may be sufficient to inhibit such an expansion [32]. As described in this study, at low water contents (8% and 12%) between the interfaces, the volume change in water content during freeze–thaw cycles is sufficient to inhibit such swelling and shrinkage due to the elasticity of pores and the W-OH concretions themselves. However, the frost heave effect is evident due to the increased water content, and the soil deterioration is further aggravated.

In addition, it should be noted that the samples in this study were prepared using a hand-held stirring mechanism, which is different from the efficient spraying method used in the field. Therefore, the mechanical properties of the samples prepared in the laboratory may be different from those produced in the field. However, the trend observed by indoor experiments should be consistent with that in the field, including the frost resistance of adding increased EVA with the increase in W-OH concentration. This paper mainly conducts a series of shear strength studies based on direct shear tests, but the research on the mechanism of interface failure is not deep enough. If conditions permit, the failure mechanism of the interface under the action of freeze–thaw cycles can be studied more in depth. At the same time, only one case of interfacial shear strength is considered in this study. To be closer to the actual engineering situation, the law of dilation and contraction of the sample during the shear test should be considered. The research results provide a solid basis for improving the freeze–thaw resistance of Pisha sandstone solidified using a W-OH solution.

## 5. Conclusions

This study investigated the interfacial shear strength between W-OH-treated Pisha sandstone and original sandstone using a direct shear test. In particular, it studies the interfacial shear strength under freeze–thaw cycles. Under W-OH concentration and freeze–thaw processes, test curves were obtained to analyze interfacial shear properties. The test results, including the shear stress-displacement curve and the peak state shear strength, are presented and analyzed. However, the undesirable durability of the original W-OH material can reduce its service life, especially under freeze–thaw cycles. Based on the mechanism, W-OH was modified to improve its freeze–thaw durability. The main conclusions of this study are as follows:(1)The samples prepared with a higher concentration of W-OH solution have better resistance to freeze–thaw cycles at the interface. At low water contents (8% and 12%) between the interfaces, the volume changes in water content during freeze–thaw cycles are sufficient to inhibit such swelling and shrinkage due to the elasticity of the pores and the W-OH concretions themselves. With the increase in interfacial water content, the effect of interfacial water content on interfacial shear strength is greater than that of the W-OH materials;(2)There is no apparent correlation between freeze–thaw cycle effects, W-OH concentration, and moisture content on the internal friction angle at the interface. The development of the freeze–thaw cycle at the interface can be attributed to the repeated solidification–melting of water. This causes the consolidation of the failure surface to fall off and gradually produce cracks, thus reducing the bonding properties of the material—the cohesion and peak strength at the interface increase with the concentration of W-OH;(3)The addition of EVA can improve the freeze–thaw durability of the interface between W-OH-treated sandstone and the original sandstone due to the improvement in the bond strength of W-OH material.

## Figures and Tables

**Figure 1 polymers-15-02131-f001:**
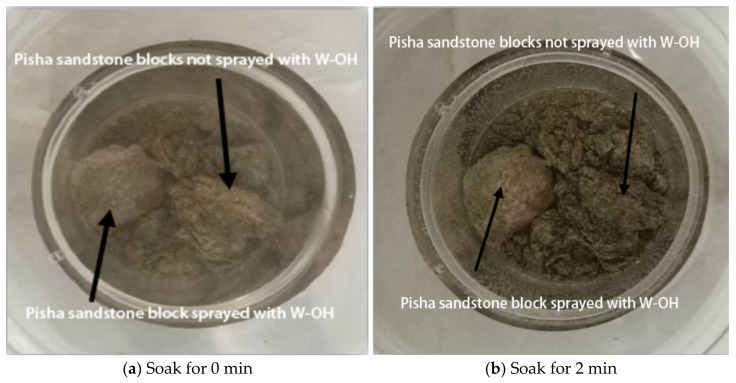
Pisha sandstone dissolves in water.

**Figure 2 polymers-15-02131-f002:**
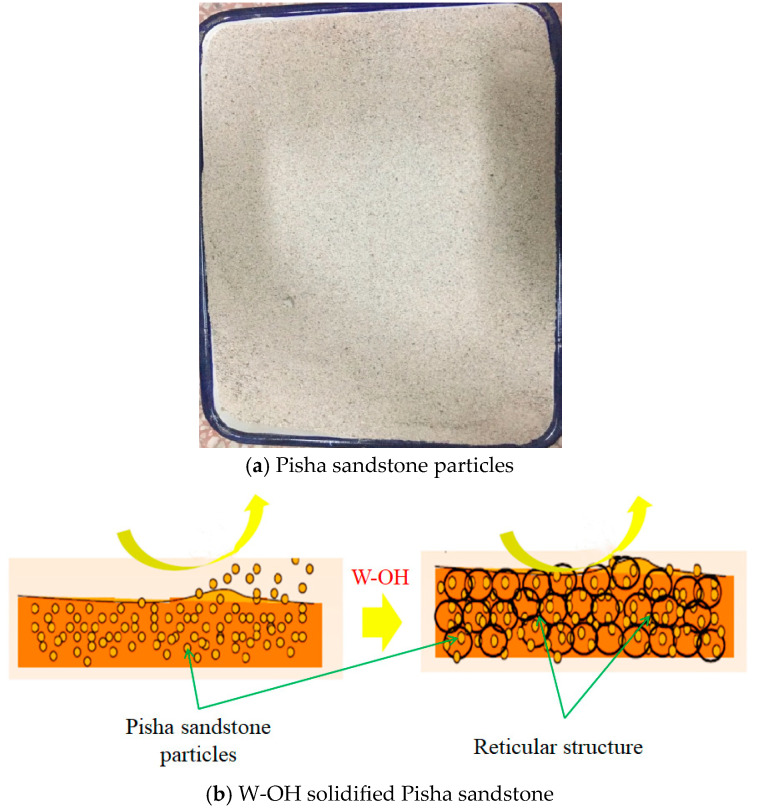
Experimental material.

**Figure 3 polymers-15-02131-f003:**
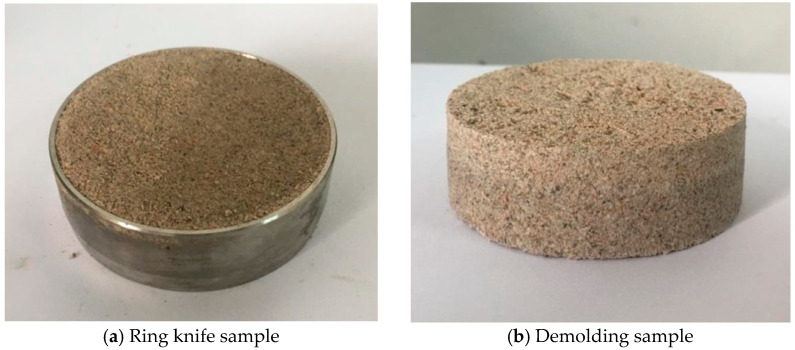
Test sample.

**Figure 4 polymers-15-02131-f004:**
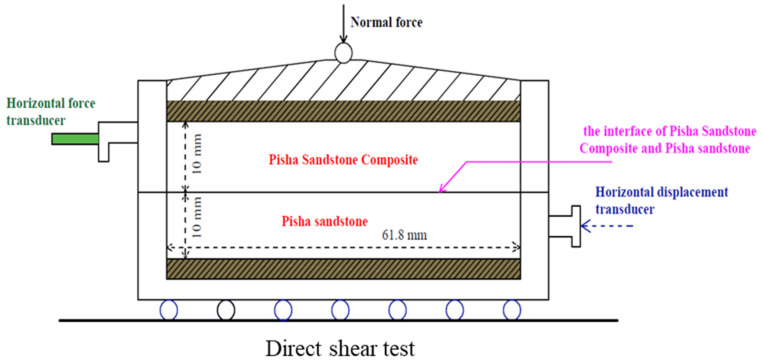
Schematic diagram of direct cutting.

**Figure 5 polymers-15-02131-f005:**
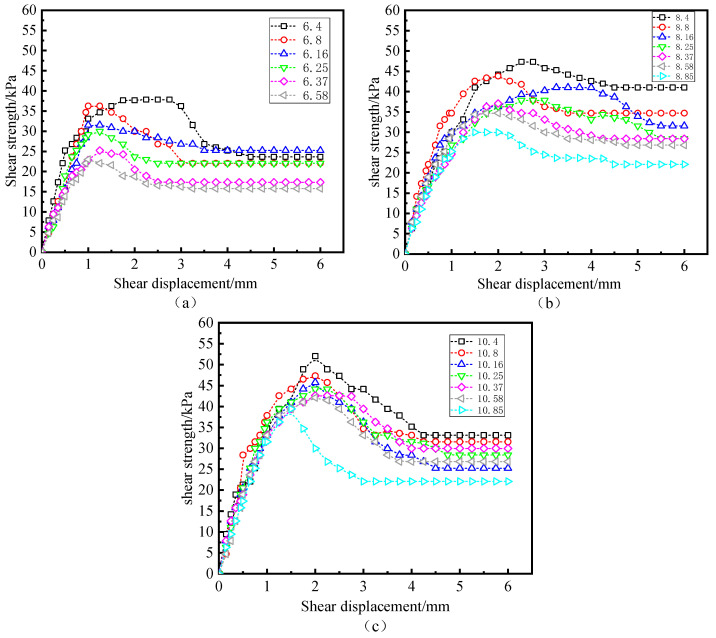
Stress-strain curve underwater content 12% and normal pressure 50 kPa (W-OH concentration of (**a**–**c**) is 6%, 8%, and 10%, respectively).

**Figure 6 polymers-15-02131-f006:**
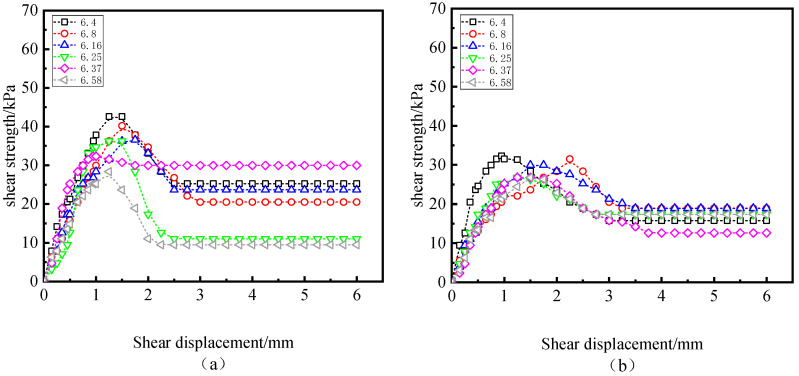
Stress–strain curves under normal pressure of 100 kPa (water content of (**a**–**c**) are 8%, 16%, and 20%, respectively).

**Figure 7 polymers-15-02131-f007:**
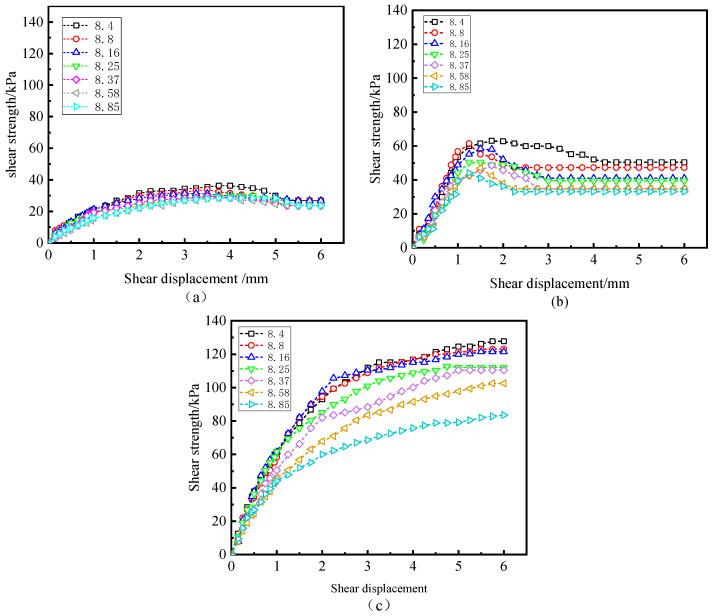
Stress–strain curve under W-OH 8% and moisture content 20% (the normal pressures of (**a**–**c**) are 50 kPa, 100 kPa, and 200 kPa, respectively).

**Figure 8 polymers-15-02131-f008:**
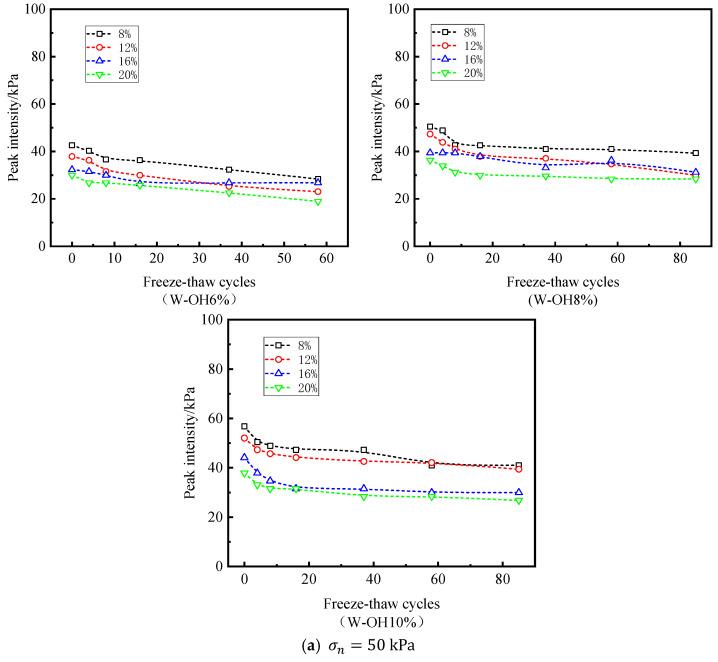
Relationship between peak intensity and freeze–thaw cycles.

**Figure 9 polymers-15-02131-f009:**
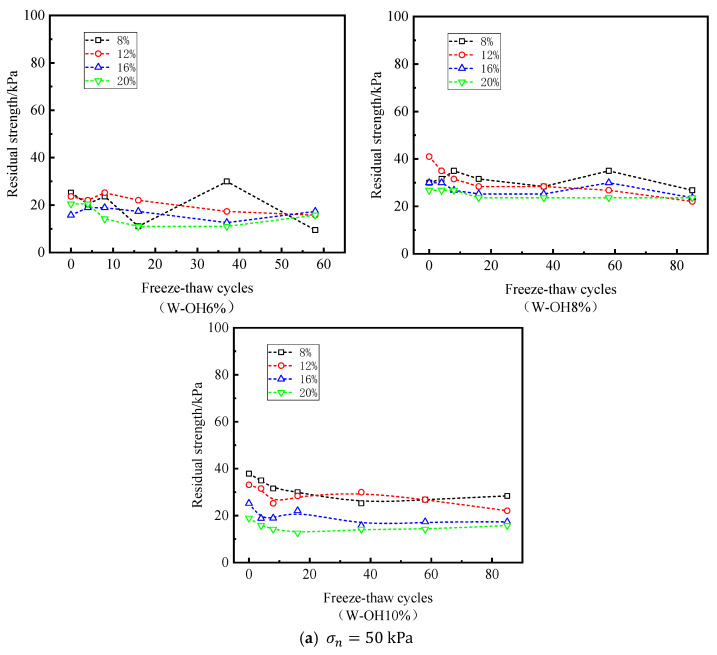
Relationship between residual strength and freeze–thaw cycles.

**Figure 10 polymers-15-02131-f010:**
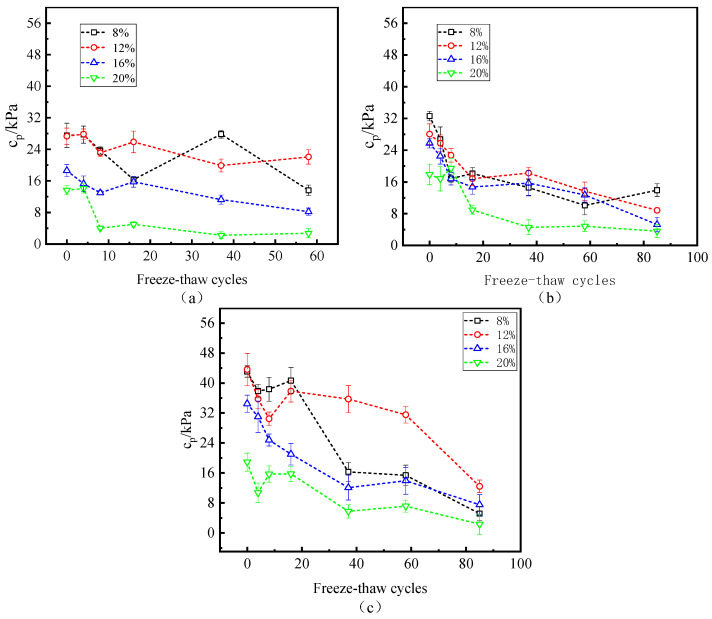
Relationship between freeze–thaw cycles and cohesion (W-OH concentrations of (**a**–**c**) are 6%, 8%, and 10%, respectively).

**Figure 11 polymers-15-02131-f011:**
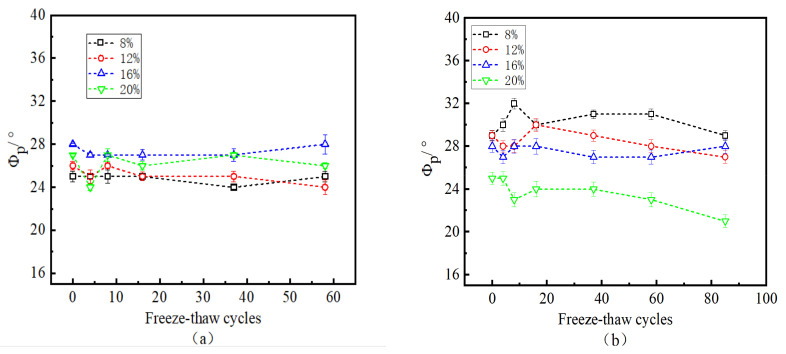
Relationship between the number of freeze–thaw cycles and the angle of internal friction (the W-OH concentrations of (**a**–**c**) are 6%, 8%, and 10%, respectively).

**Figure 12 polymers-15-02131-f012:**
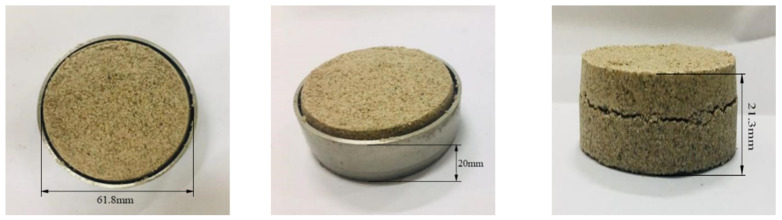
85 freeze–thaw cycles at 6% W-OH concentration.

**Figure 13 polymers-15-02131-f013:**
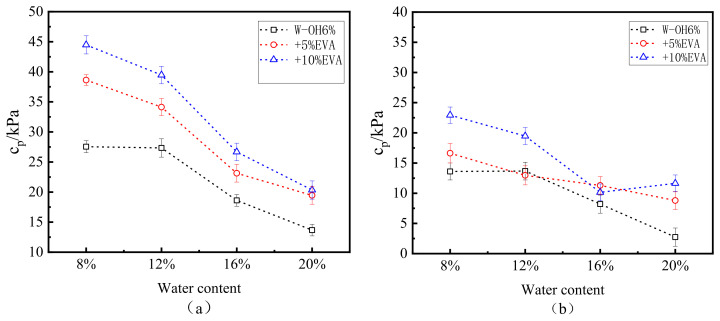
Change in cohesion of 9EVA modified W-OH under freeze–thaw cycles ((**a**,**b**) are 0 and 85 cycles, respectively).

**Figure 14 polymers-15-02131-f014:**
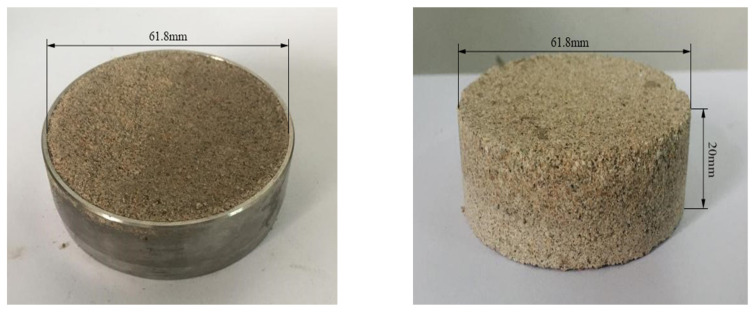
85 freeze–thaw cycles at 6% W-OH solution with EVA modification of 10%.

**Figure 15 polymers-15-02131-f015:**
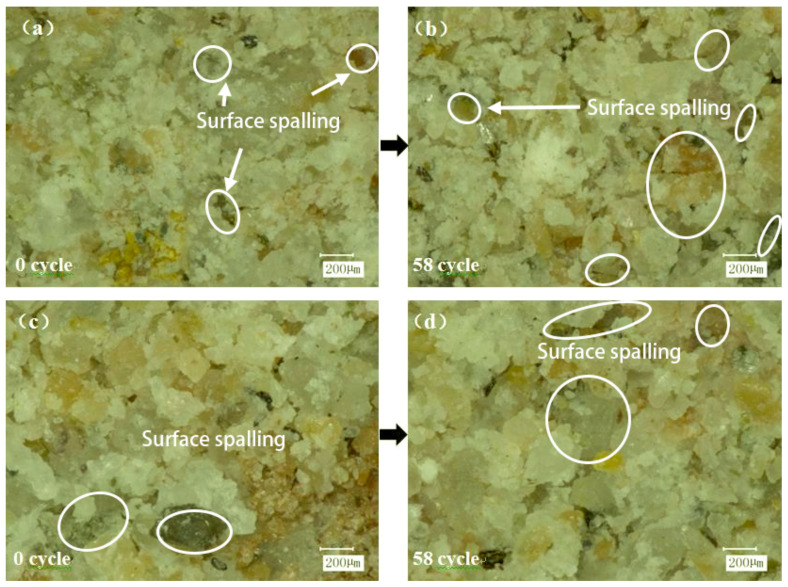
The shear failure interface of W-OH material before modification (the W-OH concentration in the figure is 6%, the moisture content at the interface (**a**,**b**) is 8%, and the moisture content at interfaces (**c**,**d**) is 20%).

**Figure 16 polymers-15-02131-f016:**
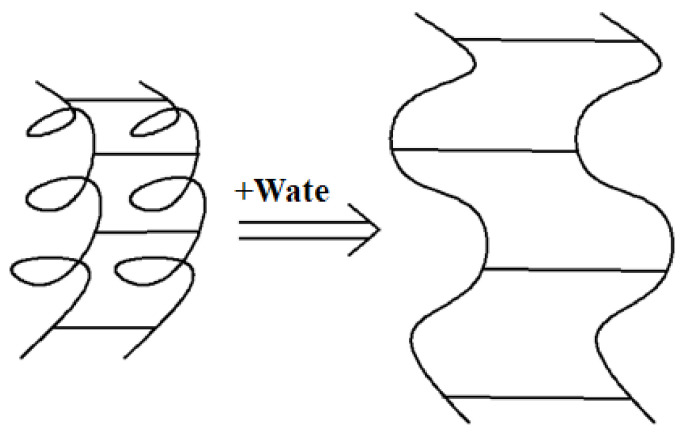
Schematic diagram of water absorption and expansion of W-OH consolidation.

**Figure 17 polymers-15-02131-f017:**
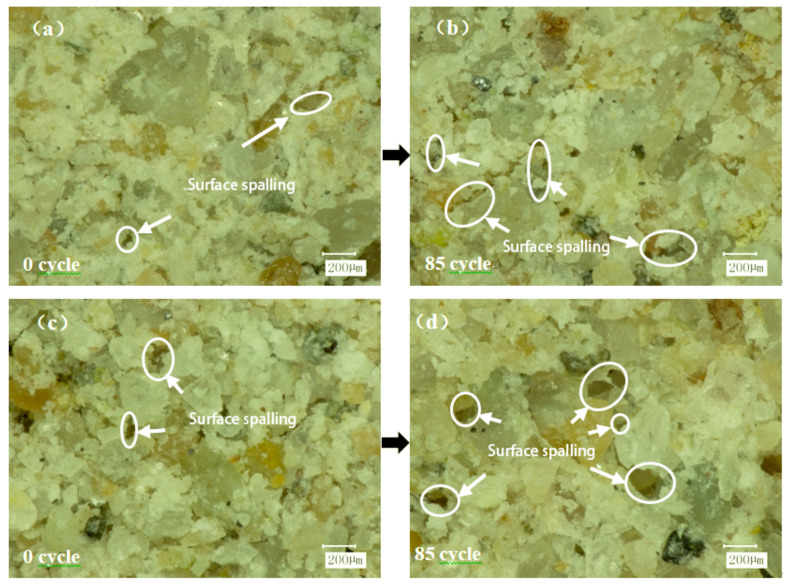
Shear failure interface of modified W-OH materials (figure W-OH materials are all modified by 6% + 10% EVA, the moisture content of interface (**a**,**b**) is 8%; the moisture content of interface (**c**,**d**) is 20%).

**Figure 18 polymers-15-02131-f018:**
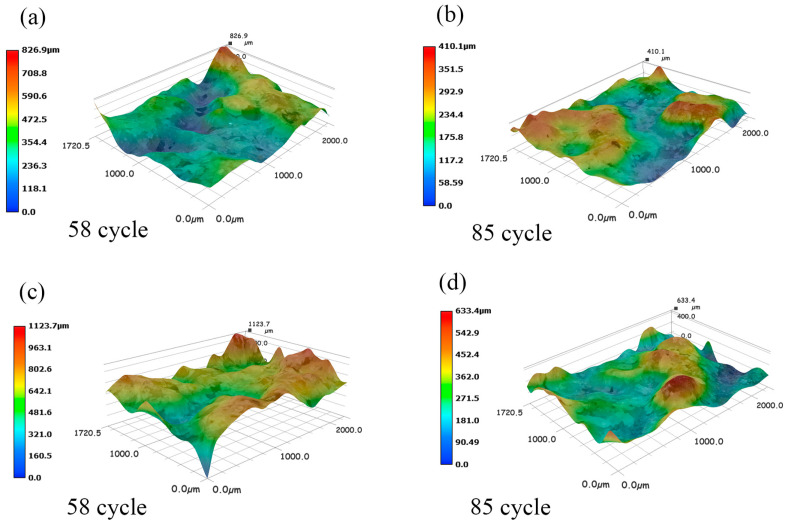
3D shape diagram of shear failure interface (6% + 10% EVA modifies W-OH materials in (**b**,**d**), the interfacial moisture content is 8% and 20%, respectively; W-OH materials in (**a**,**c**) are 6%, interfacial moisture content is 8% and 20%, respectively).

**Table 1 polymers-15-02131-t001:** Essential physical indexes of Pisha sandstone.

Water Content/%	$\mathrm{Density}/g\cdot {cm}^{-3}$	Porosity/%	Liquid Limit/%	Plasticity Index/%	Osmotic Coefficient/$mm\cdot s^{-1}$
7.70–21.1	1.85–1.96	31.04–35.15	19.6	9.40	$5.2\times 10^{-6}$

**Table 2 polymers-15-02131-t002:** Water content and solution concentration.

	Water Content/%	W-OH Concentration/%	EVA Concentration/%
Group 1	(8/12/16/20)	-	-
Group 2	(8/12/16/20)	6	-
Group 3	(8/12/16/20)	8	-
Group 4	(8/12/16/20)	10	-
Group 5	(8/12/16/20)	6	5
Group 6	(8/12/16/20)	6	10
Group 7	(8/12/16/20)	8	5
Group 8	(8/12/16/20)	8	10
Group 9	(8/12/16/20)	10	5
Group 10	(8/12/16/20)	10	10

**Table 3 polymers-15-02131-t003:** All samples of the freeze–thaw cycle.

Water	W-OH	EVA	Normal	Freeze–Thaw
Content/%	Concentration/%	Concentration/%	Stress $\sigma_{n}$	Cycles
8	6	-	50	85
12	6	-	50	85
16	6	-	50	85
20	6	-	50	85
8	6	-	100	85
12	6	-	100	85
16	6	-	100	85
20	6	-	100	85
8	6	-	200	85
12	6	-	200	85
16	6	-	200	85
20	6	-	200	85
8	8	-	50	85
12	8	-	50	85
16	8	-	50	85
20	8	-	50	85
8	8	-	100	85
12	8	-	100	85
16	8	-	100	85
20	8	-	100	85
8	8	-	200	85
12	8	-	200	85
16	8	-	200	85
20	8	-	200	85
8	10	-	50	85
12	10	-	50	85
16	10	-	50	85
20	10	-	50	85
8	10	-	100	85
12	10	-	100	85
16	10	-	100	85
20	10	-	100	85
8	10	-	200	85
12	10	-	200	85
16	10	-	200	85
20	10	-	200	85
8	6	5	50	85
12	6	5	50	85
16	6	5	50	85
20	6	5	50	85
8	6	5	100	85
12	6	5	100	85
16	6	5	100	85
20	6	5	100	85
8	6	5	200	85
12	6	5	200	85
16	6	5	200	85
20	6	5	200	85
8	6	10	50	85
12	6	10	50	85
16	6	10	50	85
20	6	10	50	85
8	6	10	100	85
12	6	10	100	85
16	6	10	100	85
20	6	10	100	85
8	6	10	200	85
12	6	10	200	85
16	6	10	200	85
20	6	10	200	85

## Data Availability

The data presented in this study are available on request from the corresponding author.

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
