# Peer review of "Effect of Hydrophilic Polyurethane on Interfacial Shear Strength of Pisha Sandstone Consolidation under Freeze–Thaw Cycles"

_polymers, 2023, doi:10.3390/polym15092131_

Round 1

Reviewer 1 Report

The manuscript, entitled "Research on Interfacial Shear Strength between Pisha Sandstone Consolidation and Pisha Sandstone Under Freeze-Thaw Cycle," presents an interesting experimental study conducted on the freeze-thaw effect on the characteristics of Pisha Sandstone. However, the introduction section is vague, and the future directions and limitations section is missing. The paper needs minor revisions before it is processed further. Some comments follow:

Abstract: The abstract must be significantly improved. The abstract is written qualitatively. The majority of the qualitative statements should be modified for quantified result comparisons.

This section must be suitable for separate presentations (independent of the manuscript text body), therefore it should include: novelty, materials and methods, and results presented in quantitative evaluation.

Introduction section.

The introduction section can be improved. The introduction presents a brief overview of the relevant literature; however, in its current form, the introduction only presents some short affirmations about the previous publication without a critical analysis (for example, line 5: "believe that long-term internal erosion will decrease soil shear strength and aggravate erosion."; this is way too vague). Please be more specific about what was previously studied, what results have been obtained, and how the current study goes beyond the state of the art.

Moreover, it seems that literature has multiple previous publications with the same aim that haven’t been cited or evaluated by the authors.

The authors state that "there are few reports on the interfacial shear properties between Pisha sandstone consolidation and Pisha sandstone at home and abroad." But no papers have been mentioned; please cite these studies and highlight the differences between this study and the previous publications.

For example, what is the difference between this study and 10.1016/j.coldregions.2020.103065?

Materials and Methods section

The lines 140–143 are redundant. The information was already presented in the first paragraph of the subsection.

Figure 4: Please remove this figure as it has no scientific value. The machine is standard; therefore, only the model number and the machine’s manufacturer are needed to assure experiment repeatability.

Experimental Results Section

Figure 13 and figure 15: please introduce a scalebar on each part of the figure.

Future directions and limitations: Please provide some future directions and limitations of the study. This section is very important for studies that propose a procedure with a limited number of experiments and no modeling of results.

Reviewer 2 Report

The article is devoted to experimental study of interfacial shear strength between hydrophilic polyurethane treated sandstone and original sandstone under conditions of alternate freezing and thawing. Hydrophilic polyurethane (W-OH) is a promising material for strengthening sandstone slopes. The article investigates the effect of W-OH content on shear strength, cohesion and angle of internal friction at different numbers of freeze-thaw cycles. The authors also proposed adding ethylene-vinyl acetate to improve the freeze-thaw durability of the interface. The article contains new interesting results. However, there are some minor remarks:

1. The authors are asked to reread paper thoroughly and correct the language. What is "everyday stress" in line 165? "Inner friction angle" instead of internal friction angle in line 258, etc.

2. Line 38: 103 instead of 103.

3. Along the y-axis in Fig. 12 the units are kPa, but the angle of internal friction is measured in degrees.

4. Line 273: Eva instead of EVA.

5. Line 355: It is stated that freezing and thawing can affect physical and chemical properties, but only mechanical and physical properties are listed below: soil shear strength, compressive strength, and soil porosity.
